# Life Cycle Assessment of PLA Products: A Systematic Literature Review

Ana Fonseca [1,*], Edgar Ramalho [1], Ana Gouveia [1], Filipa Figueiredo [1,2] and João Nunes [2]

1    Associação CECOLAB—Collaborative Laboratory towards Circular Economy,
     Rua Nossa Senhora da Conceição, nº 2, Lagares da Beira, 3405-155 Oliveira do Hospital, Portugal;
     edgar.ramalho@cecolab.pt (E.R.); ana.gouveia@cecolab.pt (A.G.); filipa.figueiredo@cecolab.pt (F.F.)
2    Associação BLC3—Campus de Tecnologia e Inovação, Centre Bio R&D Unit,
     Rua Nossa Senhora da Conceição, nº 2, 3405-155 Oliveira do Hospital, Portugal; joao.nunes@blc3.pt
*    Correspondence: ana.fonseca@cecolab.pt

**Abstract:** The rising concerns about environmental harm and pollution create a setting for the search for better materials to produce more sustainable products. Plastic plays a crucial role in modern life and most of the commonly used are of fossil origin. Polylactic Acid (PLA) has been appointed as a more sustainable alternative, due to its origins in biodegradable raw materials. This paper aims to review scientific research, where Life Cycle Assessment (LCA) is performed on this material, in order to further understand the environmental impacts and to assess whether it is a more viable option when compared to the most commonly used plastics. A systematic literature review of 81 LCA studies focused on the LCA of PLA products was conducted. An assessment of key aspects, including the system boundaries, raw materials origin, and quantitative analysis of five environmental impact categories was performed. In this comparative analysis, in addition to presenting the results for PLA products, they are also compared with other fossil-based plastics. This leads to the conclusion that PLA has higher environmental impacts on Marine Eutrophication, Freshwater Eutrophication, and Human Toxicity, which are mainly related to the agricultural phase of growing the raw materials for PLA production. For Climate Change, Polystyrene (PS) presents the higher Greenhouse Gas (GHG) emissions, and for the Ozone Layer Depletion category, Polyethylene terephthalate (PET) presents the higher impact. PLA is a solution to replace fossil plastics. However, the use of alternative biomass sources without competition with the feed and food sector could be a key option for biobased materials production, with lower environmental and socioeconomic impacts. This will be a pathway to reduce environmental impacts in categories such as climate change, marine eutrophication, and freshwater eutrophication.

**Keywords:** circular economy; eco-design; life cycle assessment; polylactic acid; review





## 1. Introduction

Plastic plays a key role in the daily life of people as well as at the industry level. It is present in almost every sector, from food storage to personal care products, and has rapidly become one of the most used materials [1]. Plastic is a particularly useful material because of properties such as lightness, malleability, flexibility, and resistance to microbes and to other natural deterioration. Recent data indicate a rising trend in plastic manufacturing and consumption, as well as waste associated with disposal. As a result of linear economic notions, plastic is extensively applied for single-use products, leading to large amounts being disposed of and persisting in the environment, causing problems in different ecosystems and human health [1].

For different ends, different types of plastic can be considered. Most of the commonly used plastics, such as polypropylene (PP), polyethylene terephthalate (PET), polystyrene (PS), polyvinyl chloride (PVC), and others, are synthetic polymers manufactured from petroleum and its allied components [2]. From production to disposal, these fossil-based

materials present a major concern. The manufacturing phase is a highly intensive energy process, which is accompanied by the release of toxic by-products, and the incorrect disposal of plastic products is a growing concern in the last decade, as the material persists in the environment, releasing macro, micro, and nanoparticles that affect almost every ecosystem [3].

A promising alternative to petro-plastics is bioplastics (also known as bio-based plastics or biopolymers), obtained from biomass such as corn, sugar cane, or lignocellulose, as part of the biorefinery concept [1]. Under suitable conditions, some types of bioplastics can either be biodegradable or compostable in a matter of months and contribute to carbon capture and storage. Bioplastics have properties similar to petro-plastics, but since they are biodegradable, they can be composted, making it less expensive to produce and complete the natural carbon cycle [4]. Among these bioplastics, polylactic acid (PLA) has been gaining traction among academic research as well as in different industries, as it can be obtained directly from renewable biological monomers, like cellulose, starch, or sucrose, by fermentation and chemical synthesis. PLA could be both biodegradable and compostable if disposed of correctly [4,5]. The majority of PLA's applications, from packing to coatings, have been for short-term products due to its mechanical characteristics and biodegradable nature [6]. PLA biocompatibility has also allowed it to advance to the forefront of applications in tissue engineering and biomedicine. Furthermore, it is a widely utilized substance for 3D printing production techniques [7].

PLA has good mechanical strength and thermal plasticity that can be used for many applications. PLA has low toughness with only 10% of elongation at the breaking point, which means PLA is a relatively brittle polymeric material [8]. The crystal structure of the PLA can be amorphous or semicrystalline in a solid state, depending on its stereochemistry and thermal history [8], and is stated to have a melting point of 151 °C with a value of heat of fusion around 21.5 J/g [8].

Although they are a viable alternative, it should be considered that the majority of bioplastics currently commercially available are produced from agricultural crop-based feedstocks (carbohydrates and plant materials), which is not optimally aligned with the United Nation's Sustainable Development Goals (SDGs), due to their competition for farmland, fresh water, and food production [1,2]. There is also a limitation for specific applications, such as food packaging, due to weak mechanical strength, brittleness, and high sensitivity to moisture [9]. It has, however, better thermal processibility than other biopolymers, such as polyhydroxyalkanoate (PHA) and polycaprolactone (PCL), besides the fact that its production needs up to less than 55% of the energy that petro-plastics require [10]. Yet, new research is being made to improve its qualities. These improvements are being achieved by means of copolymerization, composite materials, and additives, demonstrating the desire to replace petro-plastics with PLA [6].

From a circular economy perspective, based on the concept of life cycle thinking, it is possible to design products in a way that flows of by-products and waste are reintegrated into a cycle for continued exploitation, prolonging the life cycle and mitigating damages to ecosystems. For example, agriculture and other organic residues can be used as a possible alternative for the production of bioplastics, without impacting feedstocks [1,2,11].

To gain insights into the environmental effects of a material or product throughout its entire life cycle, starting from raw material acquisition and continuing until its disposal, we can use tools like life cycle assessment (LCA). The information obtained from LCA can then be incorporated into the design phase, leading to the development of Ecological Design (Eco-Design) practices [12].

### 1.1. Eco-Design

According to Directive 2009/125/EC of 21 October 2009 [13], Eco-Design is an approach to integrate environmental aspects into the development of products to improve their environmental performance throughout their life cycle. The main objective of Eco-Design is to develop products that contribute to sustainability by reducing their environ-

mental impact throughout their life cycle, along with other requirements such as functionality, quality, safety, cost, ease of production, ergonomics, and aesthetics [12].

Eco-Design can encompass other concepts such as Design for Sustainability, Design for the Environment, and Design for Recycling. Applying this concept consists of six steps outlined in ISO 14006:2020 [14]. In each stage of the life cycle, there are environmental aspects (material and energy inputs and outputs) and associated environmental impacts (such as climate change, resource depletion, toxicity, air, water, soil pollution, etc.) [15]. Eco-Design can be a tool for life cycle management since it allows the assessment of the environmental performance of products and/or services based on LCA methodology [15].

Transfer of pollution can be prevented or at least moderated via Eco-Design. Environmental performance indicators (EPI) are often used to give simplified LCA information. EPI provides data to assess the effects of actions (positive or negative) on the environment at "t" time. To develop environmental improvement targets, it is important to be well aware of the products' environmental problems as well as their context. For instance, will be important to address the environmental issues of toxic materials, substances, and types of raw materials (biomass agrocrop, residues, etc.) to define the framework of Eco-Design goals [16]. Through the examination of the results of LCA obtained for PLA, it is possible to optimize this biopolymer to make it more eco-friendly and improve its eco-design in future work [7].

### 1.2. Life Cycle Assessment

The methodology of life cycle environmental impact assessment of a given product or process was developed in the 1990s and formalized through the introduction of the norms from the International Organization for Standardization (ISO) [17].

There is a multitude of benefits in conducting LCA, such as obtaining a detailed assessment from the development of a product/ process to the strategic planning of all operations involved. On the other hand, product designers can also explore the consequences and effects of their choices regarding the ultimate sustainability of the products/processes. A correct analysis of the target impact categories can help a purchasing department to understand which suppliers have the most sustainable products/methods/processes. LCA should not be used as the only tool for decision-making but as support. In addition to theoretical LCA studies, consumers have shown great concern about purchasing sustainable products. This, in turn, encourages companies to seek more efficient and environmentally friendly methodologies [17].

There are different types of LCA. As a general rule, for a more detailed analysis, the information collected needs to be as complete as possible. For example, a report for internal use has fewer analytical requirements than a report that will be used for marketing or other external communication purposes. There are also many LCA-related assessments, such as Environmental Product Declarations (EPD) studies (in compliance with a specific product or sector standard), single-issue analyses such as carbon or water footprint, social LCA, and long-term monitoring studies. Following the standardized characteristics defined by ISO 14040:2006 and ISO 14044:2006 [18,19], LCA can be divided into four steps (Figure 1):

1.  Goal and Scope Definition:

    In this step, the context and application of the LCA study, as well as the questions to be answered by the research, are defined. System boundaries (cradle-to-gate or cradle-to-grave) and functional units (a quantifiable reference that defines the unit being analyzed) are explained in this step and should allow meaningful comparisons between different processes and products [17].

2.  Life Cycle Inventory (LCI):

    This stage involves data collection on material and energy flows throughout the entire life cycle of the defined functional unit, mainly information regarding inputs (for example, raw materials, energy, and land use) and outputs (air emissions, water, and land). The LCI

provides a detailed overview of the resource consumption and environmental emissions associated with the process [17].

3. Life Cycle Impact Assessment (LCIA):

The collected inventory data is evaluated to assess potential environmental impacts and to quantify burdens associated with the inventory flows. The impact categories, such as carbon footprint, water footprint, eutrophication, acidification, and human toxicity, are chosen to take into account the product or material in the analysis [17].

4. Life Cycle Interpretation:

The final stage of LCA is the interpretation of the results obtained from the data, to draw conclusions and make informed decisions. Life Cycle Interpretation involves evaluating the environmental performance of the defined functional units and comparing products or materials, identifying areas of improvement and trade-offs between different environmental impact categories [17].

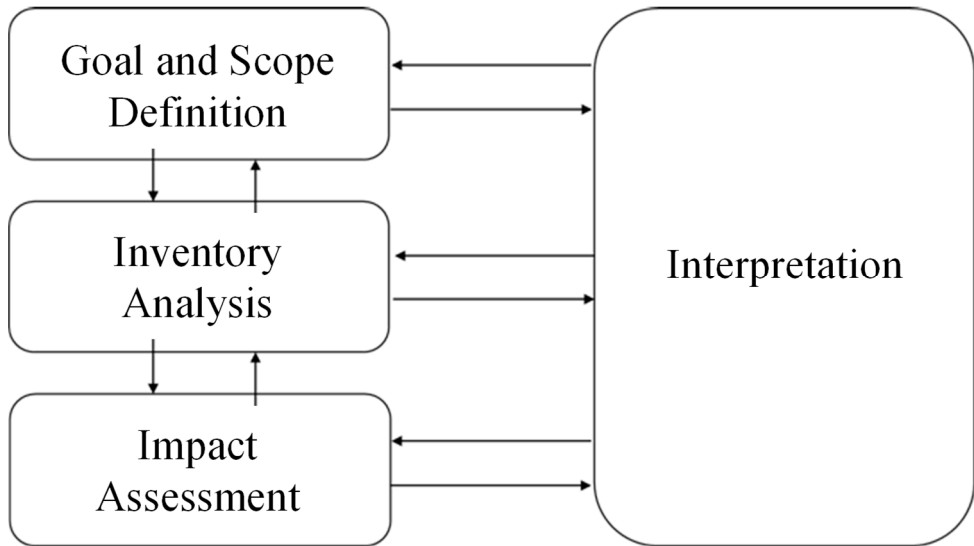

**Figure 1.** Life Cycle Assessment Methodology.

The use of LCA methodology can function as decision support in the following situations:
- Searching for pre-existing life cycles, e.g., those with the least negative impact on the environment [17];
- Making decisions in industry, public organizations, or NGOs, which can determine the direction and priorities for strategic planning, product design, or process change [17];
- Choosing indicators capable of analyzing the environmental behavior of the organization, including measurement and evaluation techniques, mainly related to the assessment of the state of the environment surrounding the system under analysis [17];
- Linking marketing with the formulation of environmental product claims or eco-labels [17].

This article describes and synthesizes the new range of LCA literature results, including studies published from 2003 to 2023, by (i) performing a qualitative assessment of the literature; (ii) quantifying the results of the following environmental impact categories: Climate Change, Ozone Layer Depletion, Freshwater Eutrophication, Marine Eutrophication, and Human Toxicity; (iii) assessing the variability in the reported results of the environmental impact categories previously mentioned. The main goals are to (i) identify the main raw materials used for the production of PLA or PLA products; (ii) quantitatively compare the results obtained by the different authors and with other types of fossil-based plastics; (iii) provide recommendations to improve the life cycle performance of new products from PLA. This article is organized into four sections, including this introduction. Section 2

presents the Materials and Methods of this systematic literature review, while Section 3 presents the main results. Section 4 draws the discussion and conclusions together.

## 2. Materials and Methods

### 2.1. Bibliographic Research

An online search of articles published between 2003 and 2023 consisting of LCA studies of PLA products was performed on Google Scholar and on Scopus using the keywords "LCA", "PLA", and" Life Cycle Assessment". The search retrieved a total of 168 papers. After a first analysis where reviews, conference papers, and book chapters were discarded, 117 papers were left, of which 27 were duplicates with the ones provided by Google Scholar, 5 were not on the theme, 3 were not written in English, and 41 were discarded after a careful abstract analysis. In the end, the total number of analyzed papers was 81, 40 from Google Scholar search, and 41 from Scopus search, as can be seen in Figure 2.

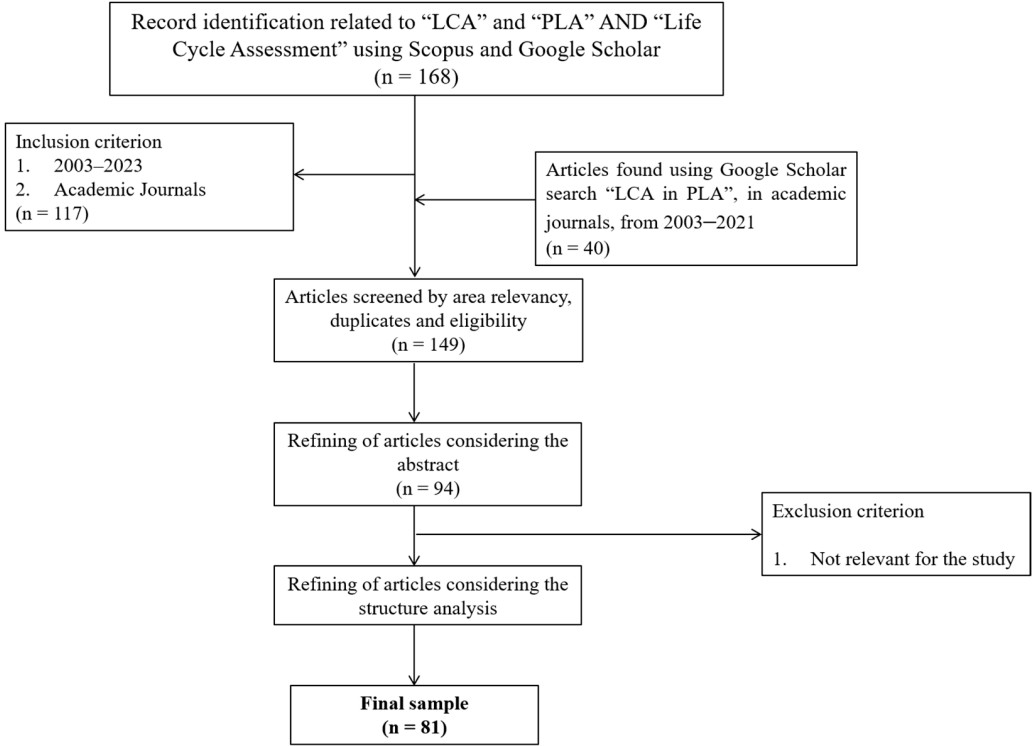

**Figure 2.** Diagram of literature search and respective screening.

Furthermore, the selection process also allowed for the identification of articles that provided a fair comparison with other polymers. To synthesize the information gathered, a table was created, highlighting the key findings that were crucial for the analysis's scope (refer to Table S1). The table was inspired by the article "The Life Cycle Assessment for Polylactic Acid (PLA) to Make It a Low-Carbon Material" by Rezvani Ghomi et al., 2021 [20].

### 2.2. Comparison of Environmental Impacts by Impact Category between Different Polymers

Impact values were collected for each environmental impact category identified for PLA and other fossil-based polymers, including Polypropylene (PP), Polyethylene Terephthalate (PET), High-Density Polyethylene (HDPE), Low-Density Polyethylene (LDPE), and Polystyrene (PS), whenever possible. Articles that conducted comparative LCA analyses between PLA and other conventional polymers derived from fossil sources, such as PP, PET, PS, LDPE, and HDPE, were given more focus, whenever those articles provided such results. To account for the heterogeneity of functional units found in the analyzed articles, conversions to a standardized unit of 1 kg of polymer were made whenever possible. The original impact categories' values were then normalized to 1 kg of polymer (through the

reference flows of each study), enabling the comparison of impacts across different articles and polymers.

## 3. Results

A total of 81 scientific articles that conducted a life cycle assessment (LCA) on the production cycle of polylactic acid (PLA) were collected. The publication date of the analyzed articles varies between the years 2003 and 2023, with greater importance being given to the most recent studies.

As seen in Figure 3, most of the analyzed articles were published in 2021 (20/81 articles). Then, the largest number of selected articles was released in 2020 (10/81) and in 2022 (9/81).

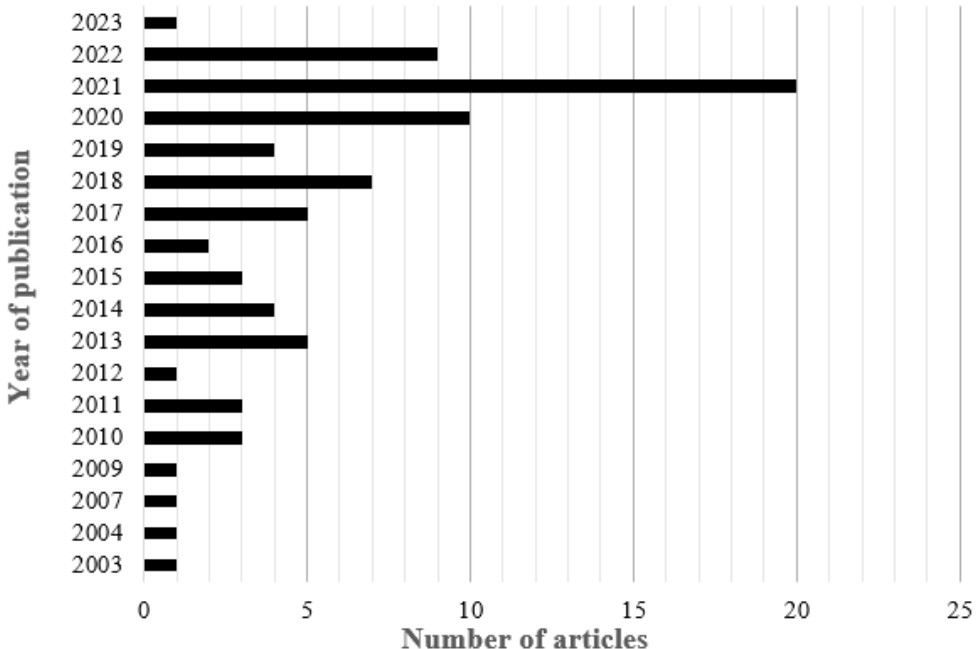

**Figure 3.** Distribution and/or frequency of the universe of the 81 articles analyzed by publication date.

In addition to the temporal aspect, it is also relevant to know the geographical borders on which the articles delimit their study, given that the stages of obtaining data and proposing scenarios mostly depend on this information. Thus, more than half of the articles studied had the European territory as their border, while the rest could be differentiated between the ones that examined the American continent and the ones that examined Asia (of these two, the first proved to be more preponderant, although with a minimal difference).

From this point on, the results will be presented depending on which stage of the LCA (explained in the Introduction) they could be associated with. It should be noted that the Life Cycle Interpretation stage is only mentioned in the discussion section.

### 3.1. Goal and Scope Definition

The objective and scope varied depending, essentially, on the focus of the analysis carried out in each of the studies. With this information, 37 of the 81 articles addressed a cradle-to-grave analysis, while 20 preferred a cradle-to-gate approach. In turn, 2 articles chose the cradle-to-cradle analysis, 2 choose the gate-to-gate, and the other 2 the gate-to-grave approach. Finally, 11 articles defined analysis objectives that were different from the ones mentioned (called "others"), while 4 did not specify which analysis was chosen ("NA"), as can be seen in Figure 4.

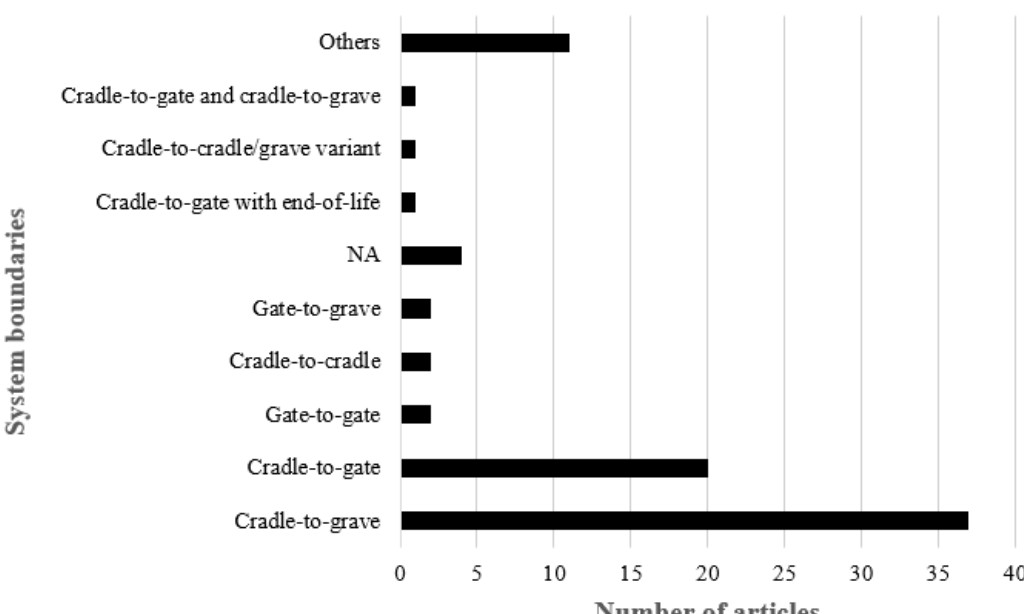

**Figure 4.** Distribution and/or frequency of the universe of the 81 articles analyzed by the objective and scope described in them.

Regarding the functional units, it was quickly noticed that the most predominant were the bottles (the most used volume being 1 L), the production of 1 kg of PLA, and the treatment of 1 ton of PLA waste produced after the industrial and the consumption stages. There were, however, some articles that did not mention which functional unit was used.

*3.2. Life Cycle Inventory*

Moving on to the inventory, most of the universe of articles analyzed (43/81) used the "SimaPro" software to support the LCA conducted. For the same purpose, the "Gabi" and "OpenLCA" software were used in, respectively, eight and five articles. However, one article used the Office for Strategy and Studies specific criteria for the lifecycle assessment of goods and services from the British Standards Institution ("PAS 2050"), while another used the software "SuperPro Designer V9.0". There were also 6 articles that, due to their analysis focus, did not apply any software ("NA") and 15 that did not mention it ("NM") (Figure 5). For the purpose of analyzing the software distributions, the version number of the software was not individually considered.

In terms of sources of information, as can be seen in Figure 6, 55 articles used "Ecoinvent" as a source of scientific data, while 36 used the existing literature on the addressed topic as a primary source of information. There were also 20 articles that were supported by industrial data and 7 that were based on the "APME—Association of Plastics Manufacturers of Europe". Finally, 23 articles used a source other than those already mentioned to substantiate their information.

It should be noted that there are articles that use more than one of these sources of information to conduct the LCA, which explains why the sum of the plots in the following figure doesn't correspond to the number of articles analyzed.

Regarding the choice of raw material for the production of PLA, 44 studies considered corn as the preferred source for this process. There were still four studies that preferred corn starch and four that chose sugar cane. On the other hand, the type of raw material was not taken into account in the LCA performed in three articles ("NA"). Finally, seven articles did not mention the type of raw material used ("NM") (Figure 7).

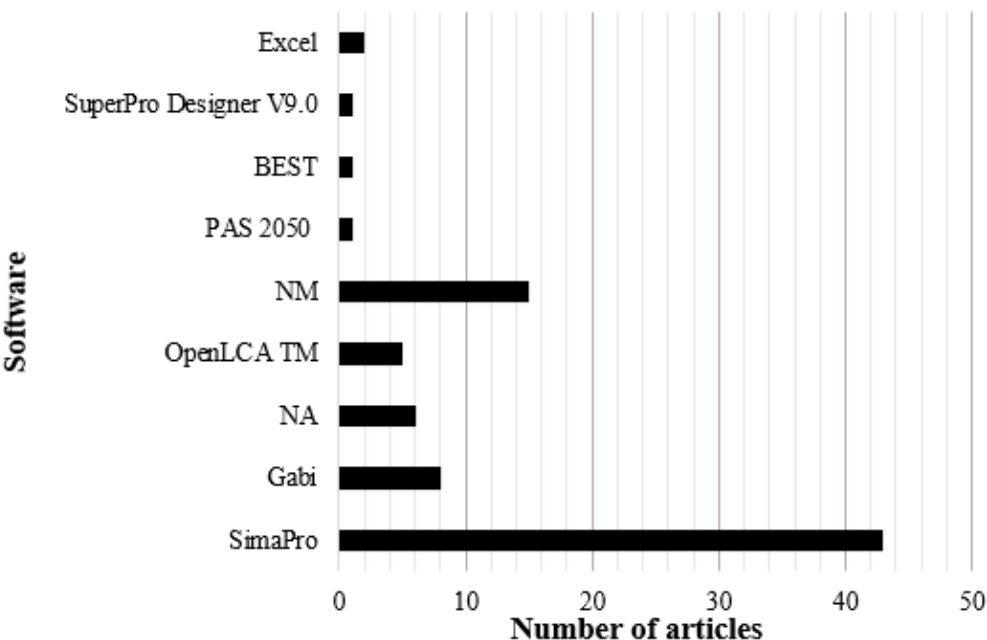

**Figure 5.** Distribution and/or frequency of the universe of the 81 articles analyzed by the type of software used.

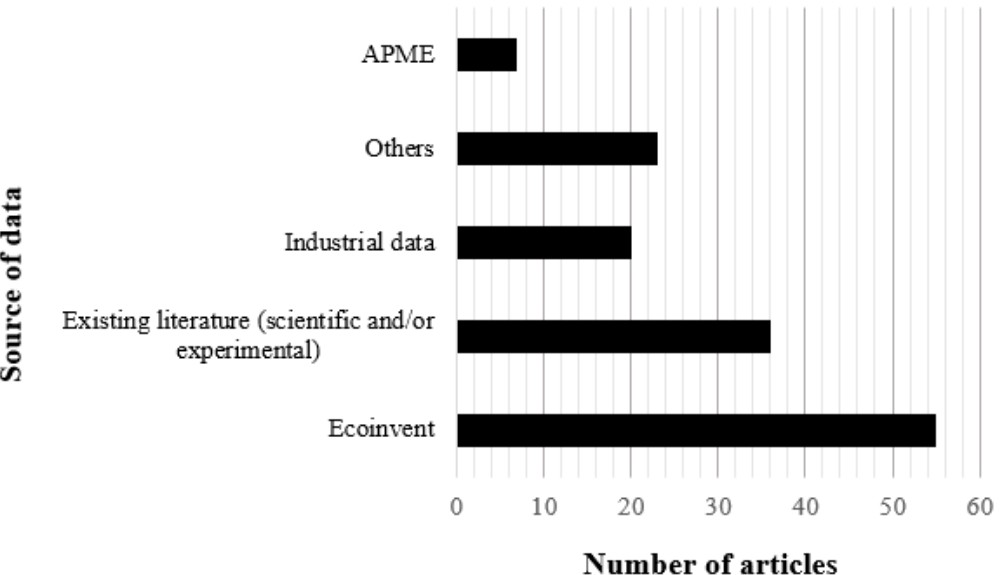

**Figure 6.** Distribution and/or frequency of the universe of the 81 articles analyzed by the sources of information used.

### 3.3. Life Cycle Impact Assessment

With regard to the impact assessment, it can be noted that the "ReCiPe" is the most used methodology, being present in 25 articles, followed by the methodology of "CML", mentioned in 11 articles.

On the other hand, "Reference Life Cycle Data System (ILCD)", "Environmental Footprint (EF)", and "IMPACT 2002+" methodologies are each used in 8, 4, and 19 studies, respectively, while the "Cumulative Energy Demand (CED)" and "Ecoindicator 99" methodologies are used by eight and six articles. Finally, these types of methodology are not applicable in 13 articles ("NA") (Figure 8).

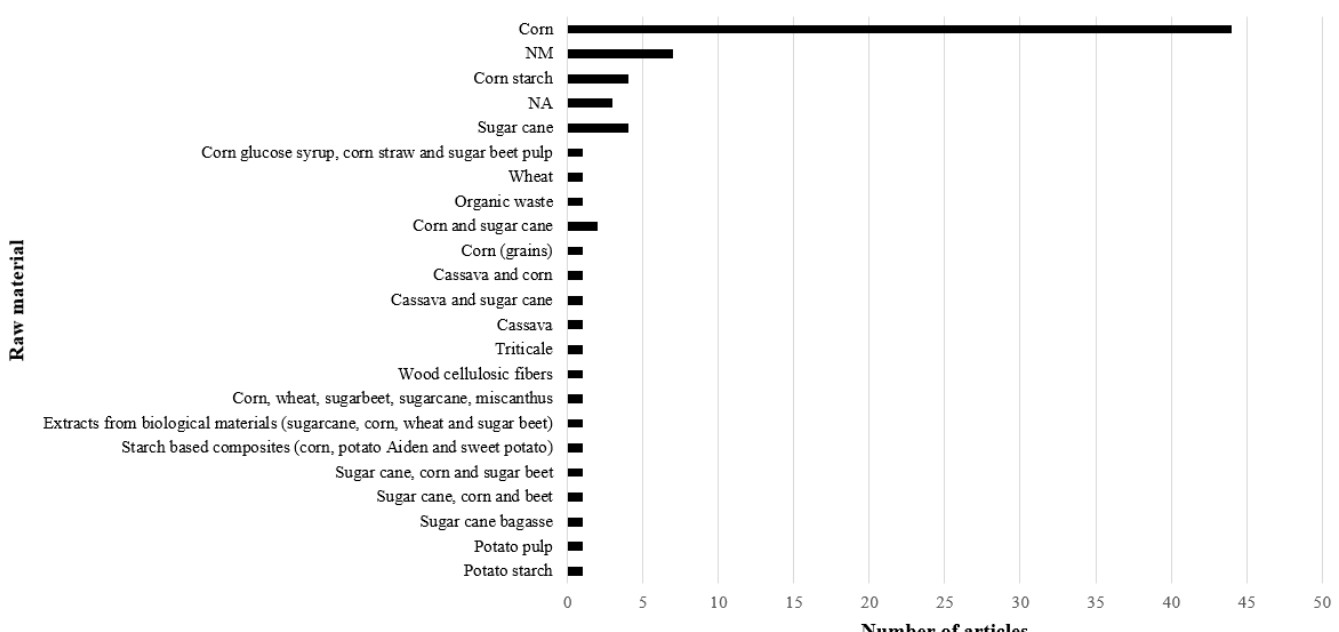

**Figure 7.** Distribution and/or frequency of the universe of the 81 articles analyzed by the different types of raw material considered.

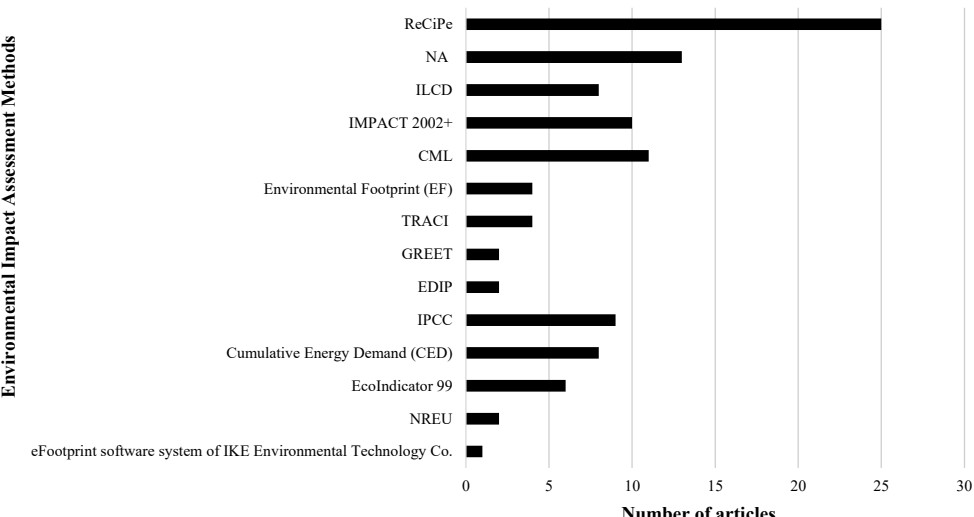

**Figure 8.** Distribution and/or frequency of the universe of the 81 articles analyzed according to the environmental assessment method used.

As seen in the software sector, there is the need to highlight that there are articles that use more than one of these methodologies to conduct the LCA, which explains why the sum of the plots in the previous figure does not correspond to the number of articles analyzed.

Afterward, impact categories were identified in the bibliography. The ones that are mentioned the most are related to the reduction of the ozone layer, global warming, climate change, and acidification. However, eutrophication of fresh and marine water, human toxicity, particulate matter formation, and land use were also highly introduced. Around the same number of articles also considered ionizing radiation, reduction of water resources, freshwater ecotoxicity, marine ecotoxicity, terrestrial ecotoxicity, photochemical formation of oxidants, non-carcinogenic agents, and reduction of fossil resources (between many others) as impact categories (Table 1).

**Table 1.** Distribution and/or frequency of the universe of the 40 articles analyzed by the impact categories identified and the different terms applied.

| Environmental Impacts | No. of Articles | Environmental Impacts | No. of Articles |
|---|---|---|---|
| Depletion of the ozone layer | 33 | Change in land use | 5 |
| Global warming | 25 | Ecosystems | 2 |
| Climate change | 22 | Carcinogenic effects on human health | 3 |
| Acidification | 24 | Water shortage | 2 |
| Fresh water eutrophication | 18 | Aquatic eutrophication | 3 |
| Marine eutrophication | 17 | Land use as an urban mesh | 2 |
| Human toxicity | 20 | Soil occupation for agriculture | 2 |
| Formation of particulate matter/ suspended particles | 16 | Resources | 2 |
| Terrestrial acidification | 15 | Use of water resources | 2 |
| Global warming potential | 31 | Fresh water acidification | 1 |
| Land use | 19 | Global warming (including and excluding biogenic carbon) | 1 |
| Ionizing radiation | 16 | Fossil fuels | 1 |
| Depletion of water resources | 10 | Depletion of mineral resources | 2 |
| Ecotoxicity in fresh water | 8 | Carbon-derived effects of biogenic origin | 1 |
| Terrestrial ecotoxicity | 16 | Non-carcinogenic effects | 1 |
| Photochemical formation of oxidants | 12 | Scarcity of fossil resources | 3 |
| Non-carcinogenic agents | 6 | Scarcity of mineral resources | 2 |
| Depletion of fossil resources | 7 | Matter in suspension | 1 |
| Fresh water ecotoxicity | 14 | Minerals | 2 |
| Marine ecotoxicity | 13 | Primary energy needs | 1 |
| Land cover | 6 | Soil and water acidification potential | 1 |
| Inorganic respiration | 11 | Ozone layer depletion potential | 4 |
| Carcinogens | 9 | Fossil fuel depletion potential | 3 |
| Abiotic depletion | 10 | Depletion potential of the aquatic environment | 1 |
| Terrestrial eutrophication | 6 | Ecotoxicity potential | 2 |
| Resource utilization | 5 | Photochemical ozone formation potential | 3 |
| Fossil fuel depletion | 6 | Soil occupation potential | 1 |
| Photochemical formation of ozone | 5 | Human toxicity potential | 4 |
| Eutrophication potential | 6 | Primary energy saving | 1 |
| Human health | 5 | Ecosystem quality | 1 |
| Aquatic ecotoxicity | 5 | Energy use (energy needs) | 1 |
| Acidification potential | 4 | Use of non-renewable energy | 7 |
| Organic respiration | 7 | Use of renewable energy | 2 |
| Smog | 3 | Use of resources (fossil fuels) | 2 |
| Use of resources (minerals and metals) | 4 | Use of resources for energy carriers | 1 |

Then, it was verified which polymers, including PLA, had more impact in the categories mentioned above. In order to compare the results collected in the articles in a more precise way, the functional unit was standardized by converting it to 1 kg of polymer, and the values of the different impact categories were adjusted accordingly.

The main impact categories described, as explained before, were "Ozone Layer Depletion", "Climate Change", "Freshwater Eutrophication", "Marine Eutrophication", and "Human Toxicity". Table 2 and Figure 9 present a summary of the average impact values of each polymer in each of the previous categories.

From the previous figure, it could be seen that PLA showed an average contribution of: (i) $7.584 \pm 1.113$ kg $CO_2$ eq/kg plastic to Climate Change; (ii) $5.831 \times 10^{-4} \pm 2.094 \times 10^{-3}$ kg CFC-11 eq/kg plastic for Ozone Layer Depletion; (iii) $1.395 \times 10^{-3} \pm 1.620 \times 10^{-3}$ kg P eq/kg plastic for Freshwater Eutrophication; (iv) $7.307 \times 10^{-3} \pm 6.189 \times 10^{-3}$ kg N eq/kg plastic for Marine Eutrophication; (v) $3.648 \pm 5.929$ kg 1,4-DB eq/kg plastic for Human Toxicity. As seen in the previous sectors, there are articles that have related more than one of these impacts to the conduction of the LCA, which explains

why the sum of the rows in the previous table does not correspond to the number of articles analyzed.

**Table 2.** Average values and standardized deviation patterns for 1 kg of polymer for the indicated environmental impact categories. The number of articles considered for these values is represented as "n".

| EIA Category for Each Plastic | PLA | PET | PP | HDPE | LDPE | PS | Refs. |
|---|---|---|---|---|---|---|---|
| OLD (Kg CFC-11 eq/kg plastic) | $5.831 \times 10^{-4} \pm$ $2.094 \times 10^{-3}$ (n = 13) | $2.668 \times 10^{-3} \pm$ $6.326 \times 10^{-3}$ (n = 7) | $1.671 \times 10^{-4} \pm$ $2.895 \times 10^{-4}$ (n = 4) | $2.161 \times 10^{-8} \pm$ $1.640 \times 10^{-8}$ (n = 3) | $3.520 \times 10^{-8} \pm$ $2.100 \times 10^{-8}$ (n = 2) | $9.660 \times 10^{-8} \pm$ $9.542 \times 10^{-8}$ (n = 2) | [1,3,11,21–32] |
| CC (Kg CO$_2$ eq/kg plastic) | $7.584 \times 10^{0} \pm$ $1.113 \times 10^{1}$ (n = 9) | $4.066 \times 10^{0} \pm$ $2.754 \times 10^{0}$ (n = 6) | $2.446 \times 10^{0} \pm$ $1.554 \times 10^{0}$ (n = 3) | $1.648 \times 10^{0}$ (n = 1) | (n = 0) | $6.134 \times 10^{0}$ (n = 1) | [1,3,21,23,25,26, 29,31–33] |
| FEW (Kg P eq/kg plastic) | $1.395 \times 10^{-3} \pm$ $1.620 \times 10^{-3}$ (n = 10) | $3.330 \times 10^{-4} \pm$ $2.538 \times 10^{-4}$ (n = 4) | $2.766 \times 10^{-5} \pm$ $1.435 \times 10^{-5}$ (n = 2) | $1.807 \times 10^{-4}$ (n = 1) | (n = 0) | $6.891 \times 10^{-5}$ (n = 1) | [3,11,22,23,26– 29,31–33] |
| ME (Kg N eq/kg plastic) | $7.307 \times 10^{-3} \pm$ $6.189 \times 10^{-3}$ (n = 10) | $2.139 \times 10^{-3} \pm$ $1.270 \times 10^{-3}$ (n = 3) | $7.777 \times 10^{-4} \pm$ $4.828 \times 10^{-4}$ (n = 2) | (n = 0) | (n = 0) | $2.101 \times 10^{-3}$ (n = 1) | [3,5,22,23,27,28, 31–34] |
| HT (Kg 1,4-DB eq/kg plastic) | $3.648 \times 10^{0} \pm$ $5.929 \times 10^{0}$ (n = 6) | $1.692 \times 10^{0} \pm$ $9.254 \times 10^{-1}$ (n = 2) | $1.069 \times 10^{-1}$ (n = 1) | $1.195 \times 10^{-1}$ (n = 1) | $1.478 \times 10^{-1}$ (n = 1) | (n = 0) | [1,11,23,30,31,33] |

EIA—Environmental Impact Assessment; CC—Climate Change; OLD—Ozone Layer Depletion; ME—Marine Eutrophication; FEW—Freshwater Eutrophication; HT—Human Toxicity.

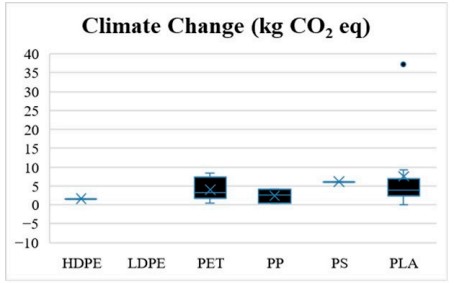

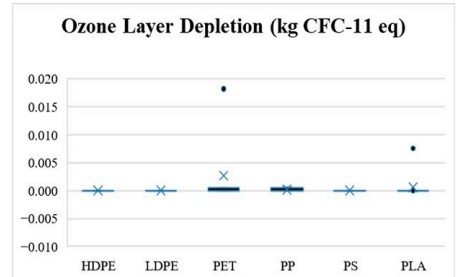

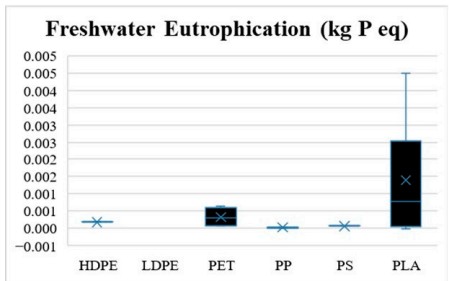

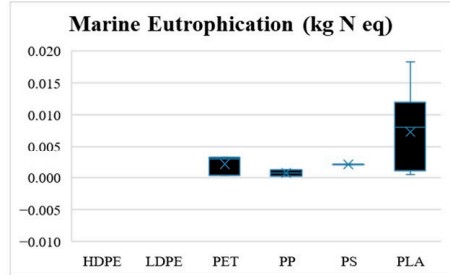

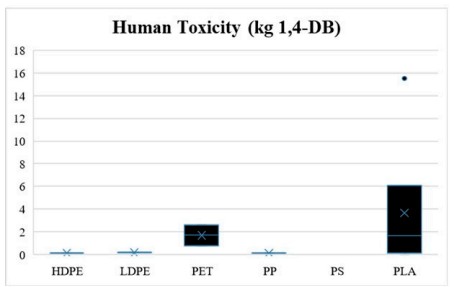

**Figure 9.** Box Plot showing median, interquartile (25–75%), and bars of maximum and minimum values of standardized environmental impact values for 1 kg of polymer, for the following impact categories: "climate change"; "ozone layer depletion"; "freshwater eutrophication"; "marine eutrophication"; "human toxicity". Legend: ●—outliers; ×—mean.

At the level of the average results of environmental impacts, the PLA presents higher environmental impacts for the following impact categories: Human Toxicity, Freshwater Eutrophication, and Marine Eutrophication. On the other hand, for the Ozone Layer Depletion category, PET is the plastic with the higher environmental impact. For Climate Change, PS presents higher GHG emissions. Since PLA and PET are the plastics for which the results have been calculated on the basis of the largest number of occurrences, they are also the most reliable for comparison. Therefore, we can conclude that with the exception of the Ozone Layer Depletion category, PLA presents the highest environmental impacts for the remaining environmental impact categories analyzed. These results are largely justified due to the high agricultural requirements associated with the cultivation of feedstock for PLA. The use and production of fertilizers combined with agricultural activities and soil tillage need to justify the results achieved for this plastic typology.

## 4. Discussion

PLA is a promising bio-based polymer that can be produced from renewable biological monomers, including second-generation feedstock, like bagasse and straw [23,35]. To try to understand the most relevant LCA impact categories related to PLA, 81 papers were chosen after a previous analysis and most of them belong to 2021 (Figure 3). This is mainly related to the importance to understand what are the most used and common categories utilized in LCA studies as of now, as it will impact future works and the ease to compare results. Naturally, papers from previous years cannot be forgotten, as using them for comparative analysis will demonstrate how LCA regarding PLA has evolved during the last two decades.

Of the 81 analyzed papers, over thirty percent have been performed with data from Europe. It is important to remember that in 2019, the Council of the European Union proposed new EU-wide rules for the 10 most single-used plastic products, commonly found in European beaches and seas. Following these new rules, many studies have since been developed to study the cycle of biodegradable plastics [36].

As seen in Figure 4, cradle-to-grave and cradle-to-gate are the two main chosen system boundaries. Tamburini et al. did a cradle-to-grave LCA evaluation of three types of bottles for drinking water, where it was concluded that polyethylene terephthalate (PET) bottles production and use had a lower impact than PLA bottles with corn-based PLA, due to the agricultural phase, and also lower than aluminum bottles, when daily washing with soap or with hot water is included [1]. Baldowska-Witos et al. did a cradle-to-gate LCA study on two alternative plastics for bottle production. The authors concluded that the production process for the PET bottles had a higher level of harmful environmental impact than the PLA bottle production process [30].

Corn-based PLA is the most studied in the literature (Figure 7). Even though it is biodegradable and compostable, depending on the origin of its raw materials, corn produced exclusively for PLA production can result in products that might be less environmentally friendly, than others made with other polymers such as PET [37]. On the other hand, PLA produced with subproducts, like sugarcane bagasse, can carry a positive environmental impact, when compared to giving other ends to this subproduct [26]. Most biopolymers found in the market currently use agricultural feedstocks, mainly maize and sugarcane, which are associated with land use change (LUC) problems [38,39], LUC is a direct reflection of the interaction between the natural environment and human activities, and how land use and the ecosystems restrict and interact with each other. Changes to LUC will have long-term environmental and ecological effects [40]. It may cancel the carbon benefits of bioplastics if all 250 million tons of plastic produced each year were switched over to bioplastics since this would require up to 5% of all arable land [39]. Currently, bioplastics are facing the same problem faced by the first generation of biofuels; while initially thought to be the solution for clean renewable fuels, soon they were linked to the "food vs fuel" theme [41]. It is, therefore, necessary to find new feedstock sources as well as to improve PLA production technology through these new sources, which could even prevent

social unrest associated with the use of bioplastics. Over the last decade, there has been an exponential increase in the use of LCA software and databases, such that, nowadays their use is state-of-the-art. Applying this system allows for a standardized procedure, direct connection to LCI databases, and depending on the software, different possibilities to visualize the results, which will aid in their interpretation [42]. Figures 5 and 6, show that the most used software is SimaPro, with GaBi, as the second most used. The main data sources are the Ecoinvent database, followed by the data collected from the literature. SimaPro has been widely used for many years for LCA studies due to its many advantages such as high flexibility, user-friendliness, and the possibility to incorporate many databases, such as Ecoinvent, as well as to introduce data manually [43]. Frequently, more than one data source is employed. Moretti et al. in their LCA study about single-use cups made from PLA, PP, and PET used a combination of primary data retrieved directly from the industry, the literature as well as the Ecoinvent database [3].

The main environmental impact assessment method used was found to be ReCiPe (Figure 8). With it being updated in 2016, it is now a state-of-the-art method that incorporates 3 endpoint impact categories (human health, ecosystem quality, and resource scarcity) and 17 midpoint categories [44]. Due to assessing several impact categories, ReCiPe method can help provide more information for future follow-up works [27].

Ozone Layer Depletion and Global Warming Potential (GWP) are the most often observed impact categories in the literature for LCA studies (Table 1). It is more accurate to say that LCA calculates potential impacts due to the inherent difficulties in accurately mapping resource utilization, emissions, and their corresponding effects, as well as the nature of aggregating impact assessments across various temporal (such as tomorrow and two decades from now) and spatial (varying global locations) dimensions [45]. Additionally, as demonstrated in Table 1, the nomenclature of effect categories lacks uniformity, highlighting the requirement for uniform titles and descriptions.

There is great heterogeneity when analyzing the functional units chosen for each work. The production of 1 kg of PLA is the functional unit most commonly selected, but there are also many papers that have chosen the production of 1000 bottles, although with different volume sizes. The disadvantage of having so many possibilities to define a function unit is that the comparison of LCA results is only valid when the two products of interest offer the same functionality for the user. This works well when the idea is to understand where through the production chain is possible to improve or which is the part with the most environmental impacts, but this creates uncertainty in the results when comparing to a new product with an improved performance of its functions or when it has additional functionalities [46].

When comparing PLA with other types of polymers, PET is the plastic most often used. PET is a biodegradable thermoplastic polymer resin with many applications, mainly in packaging and textile production, making it one of the main plastics in urban waste. Although much of it is nowadays recycled, there is still a significant part that ends up in landfills [47,48] or is discarded into the environment, making this an excellent reference material for comparisons.

Based on the analyzed articles (and on Table 2 and Figure 9) and according to the comparison made between the impacts of different plastics for the evaluated categories, PLA was the polymer with the greatest impact and contribution to the categories of "Freshwater Eutrophication", "Marine Eutrophication", and "Human Toxicity", with values of $1.395 \times 10^{-3} \pm 1.620 \times 10^{-3}$ kg P eq/kg plastic, $7.307 \times 10^{-3} \pm 6.189 \times 10^{-3}$ kg N eq/kg plastic, and $3.648 \pm 5.929$ kg 1,4-DB eq/kg plastic, respectively, followed by PET polymer. For Ozone Layer Depletion, it is the opposite; PET presents the highest impact followed by PLA. For Climate Change, PS presents the highest GHG emissions, followed by PLA and PET.

It is important to note that the presence of at least one outlier value was observed for all environmental impact categories analyzed for the PLA, other than "freshwater" and "marine eutrophication". Likewise, for the "Ozone Layer Depletion" impact category,

an outlier was also found for the PET polymer. These outliers may occur due to several factors, such as the conversion of the functional unit to 1 kg of polymer, the different methodological options used by the articles, the geographical border, and software, among other parameters.

While PLA has the most impact in some of the environmental categories, it is necessary to understand why that happens and how to change that. Some authors justify the higher impacts in some of the categories (e.g., Marine Eutrophication) due to the agricultural phase [34]. But the impacts of the agricultural phase are not only felt in "marine eutrophication". Tamburini et al. showed that the agricultural phase has a huge impact on GWP, eutrophication, human toxicity potential, ecotoxicity potential, and many others [1]. On the other hand, different ends of life for PLA will have vastly different impacts with landfilling being the worst solution [5].

With LCA to help understand how and where all these "problems" exist throughout the life cycle; it is then possible to use Eco-Design tools to mitigate them. Be it by developing new production technologies, finding new feedstocks, or discovering better ways to recycle or reutilize.

## 5. Conclusions

This paper aimed to give an overview of the state of the art of methodological characteristics of LCA in plastics, focusing on PLA, a bio-based polymer, to identify critical factors and indicators that may influence Eco-Design decisions. After compiling and analyzing the information obtained from bibliographic research, it is possible to conclude that: (1) PET bottles are frequently the object of LCA study; (2) "Simapro" is the most used software, mentioned in 52% of the studies; (3) "ReCiPe" is the most used life cycle inventory assessment method in the analyzed studies (24%), which mostly ranges from cradle-to-grave and cradle-to-gate; (4) inventories require dense information coming from variable sources, including from empirical data, literature, and database such as "Ecoinvent". Within the impact categories, "climate change" is the most frequently considered, mainly due to the prevalent comparison of the carbon footprint of PET versus PLA. However, the end-of-life management of PLA is not yet well-established, so considerations from the analyses regarding their final destination can significantly influence the results when comparing the two types of plastics, and even reverse their environmental performance. Thus, a critical view is necessary, including the possible need to add different scenarios for the future end-of-life management of PLA. For the categories for which it was possible to quantitatively identify the environmental impacts, PLA shows higher environmental impacts for Human Toxicity, Marine Eutrophication, and Freshwater Eutrophication compared to the other plastics analyzed, mainly due to the agricultural phase of raw material cultivation and the intensive use of fertilizers (most of the PLA raw material comes from agricultural crops such as corn and sugar cane). In turn, PET and PS have the highest environmental impacts for Ozone Layer Depletion and Climate Change, respectively. Therefore, LCA proves to be a valuable tool for gaining insights into the environmental performance of materials, and it can be effectively utilized in the Eco-Design process to explore alternatives to conventional fossil-based plastics if provided they demonstrate a superior environmental profile.

Developing the present study was a challenge, so it was necessary to overcome some limitations such as (i) the high number of different Functional Units (FU) found, as well as the fact that not all papers describe the reference flows, not allow to convert the environmental results from all papers; (ii) the system boundaries, some of which omit some steps, also make it harder to compare values; (iii) the different methodological options that make it impossible to compare results; (iv) the number of papers describing a LCA comparison between PLA and other plastics is also on a smaller scale, making it so that, for some categories, adding to the previously mentioned difficulties, the other polymers had a reduced number of available data or none at all; (v) the provision of quantitative results that make it possible to collate the results for each environmental impact category.

*Challenges and Future Work*

As can be seen, currently the production of PLA through food crops grown on arable land and important resource consumption (e.g., water, fertilizers, pesticides, etc.), still presents some environmental constraints compared to its fossil-based competitors. Thus, there are some challenges that we need to overcome: (i) Sufficient biomass needs to be produced while ensuring that resources are not overexploited and do not enter into direct competition with the food sector; (ii) Greenhouse gas emissions caused by biomass production and its associated land use must inevitably be reduced; (iii) Biomass production pathways must be economically competitive.

The use of alternative biomass sources—e.g., residue valorization—could be a key option for biobased materials production, with lower environmental and socioeconomic impacts, without competition with the food sector. Therefore, future work should involve:

1.  Identification of new residual sources of raw materials for the production of PLA. Another no less important aspect is related to the end of life of PLA and to the scientific validation of its biodegradable potential and potential end-of-life applications;
2.  Definition of Eco-design indicators to address environmental issues and map Eco-design strategies in order to perform an environmental assessment.

**Supplementary Materials:** The following supporting information can be downloaded at: https://www.mdpi.com/article/10.3390/su151612470/s1, Table S1. Global vision of the 81 articles analysed. Table S2. Original values for PLA used to create Table 2. Table S3. Original values of plastics used to create Table 2. References [49–105] are cited in the Supplementary Materials.

**Author Contributions:** Methodology, E.R., A.F. and A.G.; Formal analysis, E.R.; writing original draft preparation, E.R., A.F. and A.G.; writing—review and editing, E.R., A.F., A.G., F.F. and J.N.; F.F. and J.N., conceptualization, validation, visualization, resources and funding acquisition, along with the editing and supervision. All authors have read and agreed to the published version of the manuscript.

**Funding:** This research was funded by FLUI project (CENTRO-01-0247-FEDER-113565) and BeirInov project (CENTRO-01-0247-FEDER-113492) funded by European Regional Development Fund (ERDF). Filipa Figueiredo thanks her research contract funded by Interface Mission under the—Recovery and Resilience Plan (RE-C05-i02–Interface Mission–nº 01/C05-i02/2022), Collaborative Laboratories Base Fund, through CECOLAB base fund, funded by European Union NextGeneration EU. Centre Bio R&D unit | BLC3 thanks their support funded by Fundação para a Ciência e Tecnologia (FCT) UIDP/05083/2020 and UIDB/05083/2020. This work was also supported by WinBio—"Waste & Interior & Bioeconomy" (POCI-01-0246-FEDER-181335), under the Thematic Operational Programme Competitiveness and Internationalisation, COMPETE 2020, through the European Regional Development Fund (FEDER).

**Data Availability Statement:** Data is contained within the article or supplementary material.

**Acknowledgments:** We would like to thank Rita Henriques, for proofreading the English language of the text.

**Conflicts of Interest:** The authors declare no conflict of interest.

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
