# Peer review of "Life Cycle Assessment of PLA Products: A Systematic Literature Review"

_sustainability, doi:10.3390/su151612470_

Round 1
Reviewer 1 Report
Major Comments:
- The manuscript is well-written and provided a comprehensive summary of the key points from the relevant literature. However, the abstract should include highlights of the study results and the main goal of the research should be clearly stated in the introduction section. Some of my specific comments are listed below:
Title:
- The title should be clarified. Is the LCA focused only on PLA "packaging? The appendix suggests that some end products are in the form of PLA and its residues, such as films, trays, straws, lids, etc.
Abstract:
- Please highlight the major findings of the study and provide future recommendations. The abstract failed to highlight the key findings of the literature review regarding the LCA of PLA packaging.
Introduction:
- line 37: The abstract failed to highlight the key findings of the literature review regarding on the LCA of PLA packaging.
- lines 51-52: Kindly revise the statement. Claiming that PLA is the only biopolymer with biodegradable and compostable properties is too broad. PHA and other types of biopolymers (starch, chitin/chitosan, etc.) are also biodegradable and compostable. Likewise, the biodegradation of PHA is much better and more natural than PLA. PLA requires certain compost conditions (e.g. higher temperature) for it to degrade.
- Kindly indicate differences and clarify the unique aspects of the current study with existing research.
Results:
- lines 281-305: Kindly double-check the presented values.
- Table 2: Include the main sources or references of the information in the table.
Discussion:
- Kindly add discussions on the other environmental impacts not listed in Table 2. Highlight the major results related to these impacts.
Appendix:
- Include a table that tabulates the original values used, with their corresponding references, in Table 2.
Only minor English edits are required (e.g. use of punctuation).
Author Response
Dear reviewer,
We kindly thank the suggestion given to our work and send our response to each pointed topic.
Tittle:
For clarity, we will change the title with a different term for a broader range.
Abstract:
The abstract will be altered to include the highlight findings.
Introduction:
Line 51-52 – we will review the statement.
We will clarify the uniqueness of the research we conducted.
Results:
Lines 281-305 – we will double check the results
Table 2 – references will be added
Discussion:
Discussion about environmental impacts not included in table 2 will be added.
Appendix:
We will add a table with the original data.
Reviewer 2 Report
First of all, I appreciate the opportunity to the paper Life cycle assessment of PLA packaging: a systematic literature review. The paper deals with a very interesting problem.
Suggestions are below:
· The abstract is not well written. The most important findings must be emphasized.
· Keywords should include review, literature review, or SLR.
· The last paragraph in the introduction section is a short structure of the paper (several sentences for each section).
· The review paper should not just be a list of what everyone has done, but should identify trends and gaps in the literature and offer suggestions for furthering the field relative to the specific phenomenon, with a VERY STRONG CRITICAL VIEW AND VERY STRONG METHODOLOGY (see Suggested References).
· There is no SLR methodology. This kind of paper must have a very clear methodology (journals, key words, databases).
· Existing section 2 must be rewritten.
· Section 4 Discussion must be reinforced with theoretical and practical implications.
· Conclusion section is not on a satisfactory level. The conclusion in scientific papers is very important.
· Limitations of your research must be emphasized.
· Future research directions must be very strong and clear.
Suggested References
Denyer, D. & Tranfield, D., (2009). Producing a systematic review. In D. Buchanan & A. Bryman (eds.) The sage handbook of organizational research methods. Sage Publications Inc., Thousand Oaks, CA, 671-689.
Kilibarda, M., Andrejić, M., & Popović, V. (2020). Research in logistics service quality: a systematic literature review. Transport, 35 (2), 224-235.
.
Author Response
Dear reviewer,
We kindly thank the suggestion given to our work and send our response to each pointed topic.
Keywords: we will include keywords related to the term “review”
Abstract:
The abstract will be altered to include the highlight findings.
Introduction:
We will clarify the uniqueness of the research we conducted and how the paper is constructed.
Methodology:
We will clarify the methodology section and check the suggested references.
Discussion:
We will do a more complete discussion.
Conclusion:
We will revise the conclusion and include the limitations of the research and future directions.
Reviewer 3 Report
The manuscript is defiantly shown a great effort of experiments and it is worth to be published. However, I would like to address the following questions or comments to be taking in consideration when revising the manuscript:
1. It is recommended that the introduction section be strengthened. The introductory section contains little research on the advantages of Polyactive Acid (PLA) compared to traditional plastic products.
2. The introduction part is not fully cited, and a lot of new research has been carried out, for example: (1) Huan Zhang, Shuai Cao, Erol Yilmaz. Influence of 3D-printed polymer structures on dynamic splitting and crack propagation behavior of cementitious tailings backfill [J], Construction and Building Materials, 343(2022) .
3. In 1.1 section , What environmental issues can be addressed by mapping Eco-design strategies to environmental assessment indicators?
4. It is better to improve the quality of the images. The New Roman font is recommended for the text in the images and the formulae in the text.
5. In 3.1 section, it is suggested to add statistical analysis on results to know the average values and standardized deviation patterns.
6. It is better to cite more recently published papers according to the journal guideline. And authors must be correct references according to the journal guidelines.
7. It is better to have this paper extensively edited to optimise the language.
8. The conclusion must be reinforced.
In summary, the reviewer believes that this manuscript is strongly recommended for publication through the abovementioned revisions.
Minor improvement
Author Response
Dear reviewer,
We kindly thank the suggestion given to our work and send our response to each pointed topic.
- We will further include more recent work highlighting the properties and advantages of PLA.
- We will follow up on the suggested literature and revise the citations.
- Further information will be added to clarify this point.
- We will take this information into consideration.
- We do not find it relevant to add statistical analysis to section 3.1, as it accounts for the system boundaries more predominantly used in the chosen literature.
- We will confirm the references according to the journals style.
- We will double check overall language.
- We will revise the conclusion and include the limitations of the research and future directions.
Reviewer 4 Report
The manuscript titled "Life cycle assessment of PLA packaging: a systematic literature review" needs improvement in the following areas.
1. Objective of the review is unclear. Kindly mention the objective clearly in the abstract and emphasize the need for the review.
2. Introduction needs more clarity. Authors have to refer to more recent literature to have an in-depth idea about the core content of the review. Authors may refer to and cite some of the following recently published articles. https://journals.sagepub.com/doi/full/10.1177/1528083720924730, https://www.mdpi.com/2073-4360/13/11/1854, https://www.sciencedirect.com/science/article/pii/B9780128237915000156, https://www.sciencedirect.com/science/article/pii/S0921344921002792.
3. Since the PLA is analyzed for packaging applications, some important properties of PLA can also be discussed.
4. A nice graphical image portraying the stages of lifecycle analysis can be drawn.
5. Instead of simply consolidating the discussions in the conclusion section, authors may also discuss the challenges and possible solutions for the full-scale implementation of PLA in packaging applications.
6. Future scope of the study can be given as a separate section.
The manuscript needs a minimum level of fine-tuning in the English language.
Author Response
Dear reviewer,
We kindly thank the suggestion given to our work and send our response to each pointed topic.
- The abstract will be altered to highlight the findings and clarify the objectives.
- We will clarify the uniqueness of our research and how the paper is constructed.
- We will further include more recent work highlighting the properties and advantages of PLA.
- We will include a graphical image describing the life cycle stages of PLA.
- We will revise the conclusion and include the limitations of the research and future directions (this will be included in the conclusion or in a separate section such as suggested in point 6.)
Round 2
Reviewer 2 Report
Avoid bullet and numbering in Conclusions.
The paper should be accepted.
,
Reviewer 4 Report
All the comments given by the reviewer have been incorporated. The manuscript can be accepted in its current form. Best wishes to all the authors.